# Latent Optimization Variational Autoencoder for Conditional Molecule Generation

## Abstract

Variational autoencoder (VAE) is a generation algorithm, consisting of an encoder and a decoder, and the latent variable is used as the input of the decoder. VAE is widely used for image, audio and text generation tasks. In general, the training of VAE is at risk of posterior collapsing especially for long sequential data. To alleviate this, modified evidence lower bounds (ELBOs) were propsed. However, these approaches heuristically control training loss using a hyper-parameter, and are not way to solve the fundamental problem of vanilla VAE. In this paper, we propose a method to insert an optimization step of the latent variable and alternately update the encoder and decoder for maximizing ELBOs. In experiments, we applied the latent optimization VAE (LOVAE) on ZINC dataset, consisting of string representation of molecules, for the inverse molecular design. We showed that the proposed LOVAE is more stable in the training and achieves better performance than vanilla VAE in terms of ELBOs and molecular generation performance.

## 1 Introduction

Deep neural networks (DNNs) have demonstrated a dramatic performance improvement in various applications. Text extraction from image recognition, language translation, speech and natural language recognition, and personal identification by fingerprint and iris have already achieved high accuracy (Wu et al., 2016; Devlin et al., 2018; Awad, 2012; Nguyen et al., 2017). Recently, these applications became successful commercialized products.

For the purpose of generation of image, variational autoencoder (VAE) (Kingma & Welling, 2014), generative adversarial network (GAN) (Goodfellow et al., 2014), and reversible generative models (Dinh et al., 2015; 2017; Kingma & Dhariwal, 2018) were proposed and showed much progress (Bleicher et al., 2003; Phatak et al., 2009; Grathwhohl et al., 2018). These generative models were initially studied in image data and showed better performance than previous models. Since then, it has been extended in research area to generate new sentences (Iqbal & Qureshi, 2020) and to discover new drugs (Chen et al., 2018) and materials (Kim et al., 2018a).

Traditional materials research consists of four steps: molecule design, physical or chemical property prediction, molecular synthesis, and experimental evaluation. These steps are repeated until the desired molecular properties of a molecular structure are satisfied. Until now, trial-and-error techniques based on human knowledge have been widely used. However, they are time consuming and very expensive. In order to improve the traditional method, research for a high-throughput computational screening (HTCS) (Bleicher et al., 2003) was conducted. However, this also had limitations such as high computational cost, predefined molecular structures by human knowledge, and low accuracy of simulation.

Unlike the traditional approach, inverse molecular design is an attempt to find novel molecules that satisfy desired properties from exploring a large chemical space (Sanchez-Lengeling & Aspuru-Guzik, 2018). It extracts knowledge of potential molecular structures and properties from accumulated molecular structure databases (PubChem, ZINC, etc.) and proposes new molecular structures that do not exist in their database (Bolton et al., 2008; Irwin et al., 2012). With the inverse molecular design, it is possible to save cost by conducting molecular synthesis and experimental evaluation only for molecular structures having desired properties instead of searching almost infinite chemical space.

With the development of machine learning techniques, the generative models such as GAN and VAE have been applied to inverse molecular design tasks in recent years (Sanchez-Lengeling & Aspuru-Guzik, 2018; Shi et al., 2020; Yang et al., 2020; Jin et al., 2018; Wang et al., 2020). In GAN, the discriminator tries to distinguish the molecular structure from the generator, and the generator tries to generate similar molecular structure from the database, simultaneously. In VAE, encoder outputs latent variables using the molecular structure as an input, and decoder generates the original molecular structure using the latent variables from encoder.

However, there is a difficulty in training generative models, such as poor convergence. In GAN, it is difficult to learn due to the alternative update method from the two player game situation (Goodfellow et al., 2014). Problems such as the two models oscillation without adversarial, or the fact that the parameters are no longer learned in a specific situation due to the mode collapsing phenomenon, are still problems to be solved. In VAE, evidence lower bound (ELBO) for training VAE models requires training both the reconstruction and the KL-divergence loss of the latent variables, but it can cause a phenomenon called posterior collapsing. In order to prevent the posterior collapse, beta-VAE (Higgins et al., 2017), Re-balancing-VAE (Yan et al., 2019) and KL-annealing (Bowman et al., 2016) have been proposed.

In this paper, we propose a latent optimization VAE (LOVAE) that provides stable learning method by inserting the latent variable optimization technique in conditional VAE (cVAE) (Sohn et al., 2015). We apply two stages of the latent optimization to vanilla VAE. By first training encoder once, the latent variable has been optimized in the direction of reducing the training loss in the same input data. After a reparameterization of the latent variables from the updated encoder, an additional latent optimization was applied. Our proposed method, LOVAE, was compared and verified in the inverse molecular design task, and a drug-like molecular structure (ZINC dataset (Sterling & Irwin, 2015)) was used as database. We show the proposed method outperform vanilla VAE in terms of reconstruction loss and ELBO, which are training indicators. In addition, it showed a more improved appearance in generation phase. Consequently, generating molecules of LOVAE showed higher uniqueness, novelty ratio, and target property satisfaction than some of previous approaches. And, also, LOVAE generated molecules that showed higher value of penalized LogP property than the existing methods.

## 2 RELATED WORK

**Inverse molecular design** The inverse molecular design based on human-knowledge is very time consuming and relies on the intuition of the researcher, so many researchers recently tried to solve it through simulation method and others. HTCS (Bleicher et al., 2003), One of major simulation method, is an automated computational process that can rapidly identify active compounds, antibodies or genes, and the results provide starting points for many chemical works such as drug and material design (Chen et al., 2018; Kim et al., 2018a). HTCS uses a kind of brute-force method for searching and analyzing the desired chemical characteristics of molecule using a combination of hundreds or tens of millions of active compounds, but this can be disadvantage because the absolutely large amount of resources used to find the desired goals (Phatak et al., 2009). Recently, to solve the problems, artificial intelligence approaches have been widely applied in the field of the molecular design under the name of the inverse molecular design. The VAE and GAN are typical generative models and they have been applying to the inverse molecular design field (Sanchez-Lengeling & Aspuru-Guzik, 2018; Jin et al., 2020; Yang et al., 2020; Simm et al., 2020).

For the inverse molecular design, various formats for simple representation of molecular structure information have been defined instead of atom's xyz coordinates. First, MDL format that represents 3D coordinate information and binding information between adjacent atoms together. Extended Connectivity Fingerprint (ECFP) (Rogers & Hahn, 2010) and Simplified Molecular Input Line Entry System (SMILES) (Weininger et al., 1989) are another representation for the molecular structure as a sequential character string. And recently, the method of representing the molecular structure as a graph structure is also being researched. Among them, SMILES, string representation of molecules, is relatively easy to handle and has been showing a good performance.

**Generative models** The generative models such as GAN and VAE are very sensitive to the latent variables. In order words, the training of latent variables greatly affects the performance of the generative models. However, according to way of dealing with latent variables, difficult problems

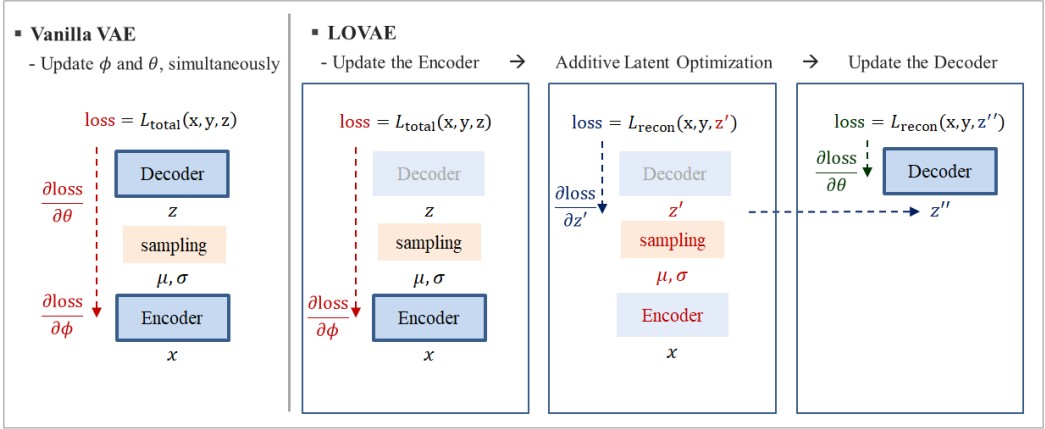

Figure 1: Comparison of the vanilla (conditional) VAE and LOVAE

have been occurred such as posterior collapsing and mode collapsing in the training. As a solution for this phenomenon, methods to adjust a KL loss weight to a value other than 1 have been proposed in VAE. In Kim et al. (2018b), latent variable from encoder is optimized according to maximizing of ELBO, and both of the encoder and decoder parameters are updated as the optimized latent variable. In Zhang et al. (2020), the reconstruction loss on the latent space is additive to the previous loss term. In GAN, there are several works applying the latent variable optimization. The method in Bojanowski et al. (2017) optimizes the latent variable by assuming that the latent variable is learnable noise. In Wu et al. (2019), randomly sampled latent variables are performed a gradient descent in the direction of reducing the loss of GAN. After that, the parameters of the discriminator and generator are updated with the loss from the optimized latent variable $z$. All of these approaches resulted in a training stability and performance improvement in GAN.

## 3 CONDITIONAL VAE

In the conditional VAE (Kang & Cho, 2018; Kingma et al., 2014), the input variable $\mathbf{x}$ is assumed to be generated from a generative distribution $p_\theta(\mathbf{x}|\mathbf{y}, \mathbf{z})$ conditioned on the output variable $\mathbf{y}$ and latent variable $\mathbf{z}$. The prior distribution of $\mathbf{z}$ is assumed to be $p(\mathbf{z}) = \mathcal{N}(\mathbf{z}|\mathbf{0}, \mathbf{I})$. We use variational inference to approximate the posterior distribution of $\mathbf{z}$ given $\mathbf{x}$ and $\mathbf{y}$ by

$$q_\phi(\mathbf{z}|\mathbf{x}, \mathbf{y}) = \mathcal{N}(\mathbf{z}|\boldsymbol{\mu}_\phi(\mathbf{x}, \mathbf{y}), \mathrm{diag}(\boldsymbol{\sigma}_\phi(\mathbf{x}, \mathbf{y}))). \tag{1}$$

From the perspective of the auto-encoder, $q_\phi(\mathbf{z}|\mathbf{x}, \mathbf{y})$ and $p_\theta(\mathbf{x}|\mathbf{y}, \mathbf{z})$ are called as an encoder and a decoder, respectively. Feed-forward neural networks are used for $\boldsymbol{\mu}_\phi(\mathbf{x}, \mathbf{y})$ and $\boldsymbol{\sigma}_\phi(\mathbf{x}, \mathbf{y})$.

The objective of the conditional VAE is to maximize ELBO, which is a lower bound of the marginal log-likelihood:

$$\log p_\theta(\mathbf{x}, \mathbf{y}) \geq \mathbb{E}_{q_\phi(\mathbf{z}|\mathbf{x}, \mathbf{y})} \log p_\theta(\mathbf{x}|\mathbf{y}, \mathbf{z}) - \mathrm{KL}(q_\phi(\cdot|\mathbf{x}, \mathbf{y})||p(\cdot)), \tag{2}$$

up to an additive constant, where KL denotes the Kullback-Leibler divergence. Given a random sample $\mathbf{z}$ generated from the encoder $q_\phi(\mathbf{z}|\mathbf{x}, \mathbf{y})$, define the total loss as

$$\mathcal{L}_{\mathrm{total}}(\mathbf{x}, \mathbf{y}, \mathbf{z}) = -\log p_\theta(\mathbf{x}|\mathbf{y}, \mathbf{z}) + \mathrm{KL}(q_\phi(\cdot|\mathbf{x}, \mathbf{y})||p(\cdot)). \tag{3}$$

Then, $-\mathcal{L}_{\mathrm{total}}(\mathbf{x}, \mathbf{y}, \mathbf{z})$ is Monte Carlo approximation of ELBO equation 2. We define $\mathcal{L}_{\mathrm{recon}}(\mathbf{x}, \mathbf{y}, \mathbf{z}) = -\log p_\theta(\mathbf{x}|\mathbf{y}, \mathbf{z})$, because it can be regarded as a reconstruction loss. In vanilla conditional VAE, parameters $\theta$ and $\phi$ are jointly optimized to minimize $\mathcal{L}_{\mathrm{total}}$.

A string representation of a molecule called SMILES is widely used to analyze molecular data (Higgins et al., 2017; Yan et al., 2019; Kang & Cho, 2018). To deal with string data like SMILES in conditional VAE, recurrent neural networks (Yan et al., 2019; Kang & Cho, 2018) are used for the decoder $p_\theta(\mathbf{x}|\mathbf{y}, \mathbf{z})$. Given a target molecular property $\mathbf{y}$, a new molecule $\mathbf{x}$ having this property is generated in the following way:

$$\mathbf{z} \sim p(\mathbf{z}), \ \mathbf{x} \sim p_\theta(\mathbf{x}|\mathbf{y}, \mathbf{z}). \tag{4}$$

---

**Algorithm 1** An update of the encoder and decoder in LOVAE

---

# Update the Encoder
Generate $\mathbf{z} \sim q_\phi(\mathbf{z}|\mathbf{x}, \mathbf{y})$.
Calculate $\mathcal{L}_{\text{total}}(\mathbf{x}, \mathbf{y}, \mathbf{z})$.
Update $\phi' \leftarrow \phi$ using $\mathcal{L}_{\text{total}}(\mathbf{x}, \mathbf{y}, \mathbf{z})$.

# Additive Latent Optimization
Generate $\mathbf{z}' \sim q_{\phi'}(\mathbf{z}'|\mathbf{x}, \mathbf{y})$.
Update $\mathbf{z}'' \leftarrow \mathbf{z}' - \frac{\alpha}{\beta + ||g||^2} g$ where $g = \frac{\partial \mathcal{L}_{\text{recon}}(\mathbf{x}, \mathbf{y}, \mathbf{z}')}{\partial \mathbf{z}'}$.

# Update the Decoder
Calculate $\mathcal{L}_{\text{recon}}(\mathbf{x}, \mathbf{y}, \mathbf{z}'')$.
Update $\theta' \leftarrow \theta$ using $\mathcal{L}_{\text{recon}}(\mathbf{x}, \mathbf{y}, \mathbf{z}'')$.

---

# 4 LATENT OPTIMIZATION VAE

At first, we have considered advancing VAE in terms of a latent variable optimization. The problem we originally wanted to consider in this paper is that vanilla VAE training is not in an optimized process. The decoder is trained depending on the encoder output $\mathbf{z}$. However, in the vanilla VAE, the latent value, which is the result of encoder before the update with the same input data $\mathbf{x}$, is used for the decoder training. From the perspective of the decoder, when the same input data $\mathbf{x}$ is used, it may be more effective to calculate $\mathcal{L}_{\text{total}}$ by using the updated latent variable ($\mathbf{z}'$) passing through the updated encoder. Our proposed method, LOVAE, tried to solve this problem in terms of the latent variable optimization. LOVAE uses the same input $\mathbf{x}$ for learning the encoder and decoder, and ($\mathbf{z}'$) is used for updating the decoder because the encoder is updated first. From this approach, $\mathcal{L}_{\text{total}}$ becomes smaller than the vanilla VAE ($\mathcal{L}_{\text{total}}(\mathbf{x}, \mathbf{y}, \mathbf{z}) > \mathcal{L}_{\text{total}}(\mathbf{x}, \mathbf{y}, \mathbf{z}')$). In addition, it helped the decoder training by optimizing ($\mathbf{z}'$) in the direction of reducing the $l\mathcal{L}_{\text{total}}(\mathbf{x}, \mathbf{y}, \mathbf{z}')$ one more in a way that does not spoil the training of encoder and decoder. That is, better encoder and better latent variable can make the decoder even better.

To be more specific, the encoder is updated as usual with the decoder while the decoder is fixed, and optimization of the latent variable from the encoder follows. Finally, the decoder is updated with the optimized latent variable. Updating encoder first has an effect similar to latent optimization. By this updating, a more suitable $\mathbf{z}$ can be created and this latent variable not only reduces the loss but also depends on the input data. A brief comparison of the vanilla (conditional) VAE and LOVAE is described in Figure 1.

First, the encoder parameter $\phi$ is updated to $\phi'$ in the direction of reducing the total loss $\mathcal{L}_{\text{total}}(\mathbf{x}, \mathbf{y}, \mathbf{z})$ where $\mathbf{z}$ is generated using the current encoder parameter $\phi$. Secondly, $z'$ is generated using the updated encoder parameter $\phi'$, and then $z'$ is updated to $z''$ in the direction of reducing the reconstruction loss $\mathcal{L}_{\text{recon}}(\mathbf{x}, \mathbf{y}, \mathbf{z}')$ using the natural gradient descent method (Wu et al., 2019). Lastly, the decoder parameter $\theta$ is updated to $\theta'$ in the direction of reducing the reconstruction loss $\mathcal{L}_{\text{recon}}(\mathbf{x}, \mathbf{y}, \mathbf{z}'')$ using the optimized $z''$.

Since the optimized $z''$ is used in the update of the decoder parameter, LOVAE is expected to achieve bigger ELBO and show stable convergence. This will be verified with numerical results in Section 5. Note that latent optimization is only applied in the training, and the inference of LOVAE remains same with the vanilla one.

In summary, the whole process of an update is detailed in Algorithm 1.

# 5 EXPERIMENTAL RESULTS

## 5.1 EXPERIMENT SETUP

ZINC database (Sterling & Irwin, 2015) is a database that organizes information about various compounds drug-like molecules. ZINC contains 3D structural information of compound quality and molecular physical properties such as molecular weight (molWt), partition coefficient (LogP),

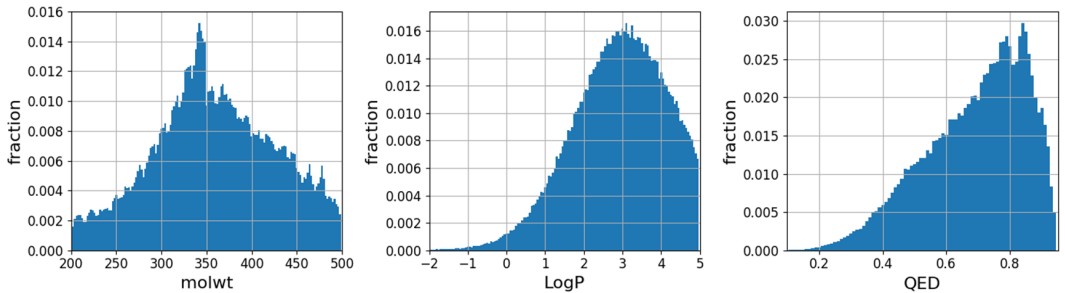

Figure 2: Example of SMILES in ZINC dataset: COc1ccc(N2CC(C(=O)Oc3cc(C)ccc3C)CC2=O)cc1

Figure 3: Distribution of target properties

and quantitative estimation of drug-likeness (QED). In addition, this information is provided in conformity with various molecular expression formats such as SMILES, mol2, and 3D SDF used in the chemical field. SMILES is a string representation to express chemical substances, such as molecules, in the form of ASCII string. It is possible to express complex graph-structured chemicals in the sequence form using simple rules. Figure 2 shows SMILES of a molecule in ZINC DB, and we used two types of DB, ZINC310K (Yan et al., 2019; Kang & Cho, 2018) and ZINC250K (Kusner et al., 2017). The vocabulary contains 39 different symbols {1, 2, 3, 4, 5, 6, 7, 8, 9, +, -, =, #, (, ), [, ], H, B, C, N, O, F, Si, P, S, Cl, Br, Sn, I, c, n, o, p, s, \, /, @, @@}. The minimum, median, and maximum lengths of a SMILES string of ZINC310K are 8, 42, and 86, respectively. (In case of ZINC250K, 9, 44, and 120)

The distribution of target properties in our sampled dataset, ZINC310K, is shown in Figure 3. Median values of molwt, LogP, and QED are 359.02, 2.91, and 0.70, respectively.

Among the existing VAE variants, beta-VAE (Higgins et al., 2017), re-balancing VAE (Yan et al., 2019), and KL-annealing VAE (Bowman et al., 2016) control the weight of KL loss in the total loss equation 3 to achieve their own purpose, such as the disentanglement of latent variables, the avoidance of the posterior collapse, and the stability of the training. The proposed latent optimization technique is also applicable with those methods.

Unlike LOVAE, semi-amortized VAE (SA-VAE) in Kim et al. (2018b) updates latent variable without the encoder update phase and applied a momentum based optimization multiple times. After that, decoder and encoder parameters are updated according to the optimized latent variable. According to (Kim et al., 2018b), SA-VAE utilized the encoder and latent optimization parts even in test phase. That is, SA-VAE is not proper to our task, because it needs the latent optimization part. On the other hand, LOVAE does not utilize the additive latent optimization in the inference phase.

Like LOVAE, there is an existing study that learns the encoder first (He et al., 2019). In the case of He et al. (2019), the encoder is updated several times until certain condition is satisfied, and different input data are used when updating encoder and decoder. LOVAE differs in some ways. It updates the encoder first, but updates it only once without any stop condition, and uses the same input data for the encoder and decoder, and performs the additive latent optimization with the reparameterization to help overall VAE learning. We think $\mathbf{z}'$ obtained by the updating encoder at the same input data is more natural and suitable for the decoder learning. In addition, we think that using the same input data for the training encoder and decoder is more effective in terms of the latent optimization than using different input data to the encoder and decoder. Also, Table 1 shows that LOVAE requires less training time. We referred to the experimental results of Kim et al. (2018b). Although SA-VAE and

Table 1: Comparison of total training time, in terms of relative speed

|                | vanilla VAE | LOVAE | He et al. (2019) | SA-VAE |
|----------------|-------------|-------|------------------|--------|
| Relative speed | **1.00**    | **1.75** | 2.20          | 9.91   |

Table 2: Training loss of ZINC310K and ZINC250K

| | **vanilla VAE** | | | **LOVAE** | | |
|---|---|---|---|---|---|---|
| **ZINC310K** | recon. loss | KL loss | total loss | recon. loss | KL loss | total loss |
| original ELBO | 20.86 | 4.38 | 25.23 | **17.30** | 6.51 | **23.80** |
| reduced KL term | 14.90 | 9.31 | 24.21 | **7.72** | 15.35 | **23.07** |
| | **vanilla VAE** | | | **LOVAE** | | |
| **ZINC250K** | recon. loss | KL loss | total loss | recon. loss | KL loss | total loss |
| original ELBO | 16.69 | 6.13 | 22.81 | **14.76** | 7.34 | **22.10** |
| reduced KL term | **5.93** | 16.63 | 22.55 | 8.99 | 12.52 | **21.50** |

(He et al., 2019) used Yahoo dataset from Yang et al. (2017), it was compared with the same method, vanilla VAE.

The basic model structure in this paper follows the general VAE structure for sequential data. The encoder has a bi-directional RNN structure, and the decoder is a uni-directional RNN structure (Yan et al., 2019; Kang & Cho, 2018). Each RNN structure was composed of three layers of GRU cells, and the dimension of the latent variable was set to 100. The hidden size of each GRU cells is 250, and the dimension of the properties is 3. A 103-dimensional vector in which a 100-dimensional latent variable and a 3-dimensional property are concatenated is inputted to the decoder. Vanilla VAE and LOVAE have the exactly same model structure, but only training strategy differs.

We used the Adam optimizer with $\beta_1 = 0.9$, $\beta_2 = 0.999$, and $\epsilon = 10^{-6}$, and a polynomial-based learning rate decay was applied. The initial and end learning rate for the learning rate decay were 0.001 and 0.0, respectively. In the case of max epoch for the training, several cases were tried. Among them, we set a number of the max epoch that shows good performance. During the training, we normalize each property value to have a mean 0 and standard deviation 1. For the additive latent optimization of LOVAE, $\alpha = 0.001$ and $\beta = 5$ were applied.

## 5.2 EVALUATION: VAE TRAINING PHASE

As defined in Section 3, $\mathcal{L}_{\text{total}}$ of VAE is the sum of $\mathcal{L}_{\text{recon}}$ and $\mathcal{L}_{\text{KL}}$. It can be thought that $\mathcal{L}_{\text{total}}$ is primarily important, and the importance of $\mathcal{L}_{\text{recon}}$ and $\mathcal{L}_{\text{KL}}$ can be determined depending on the purpose. For the evaluation, models initialized with 5 different random seeds were trained in each algorithm, vanilla VAE and LOVAE. The vanilla VAE is based on cVAE. The trained models were compared and analyzed with $\mathcal{L}_{\text{total}}$ and $\mathcal{L}_{\text{recon}}$ of the train set. The loss of LOVAE was measured without the additive latent optimization. Table 2 shows the training results for vanilla VAE and LOVAE. We tried each method five times and calculated the mean of the training losses. 'Original ELBO' utilized equation 3 as the total loss, and 'reduced KL term' reduced the weight of $\mathcal{L}_{\text{KL}}$ of equation 3 like Yan et al. (2019). In our experiments, when the weight of KL loss was from 0.7 to 0.8, each method showed a smaller $\mathcal{L}_{\text{total}}$. It can be seen that LOVAE is better than vanilla VAE in both $\mathcal{L}_{\text{recon}}$ and $\mathcal{L}_{\text{total}}$. In the results of 'original ELBO', $\mathcal{L}_{\text{recon}}$ of LOVAE is on average 2.74 less than that of vanilla VAE. In terms of $\mathcal{L}_{\text{KL}}$, LOVAE is 1.68 greater, but $\mathcal{L}_{\text{total}}$ of LOVAE is 1.07 smaller. In case of 'reduced KL term', LOVAE also shows a smaller $\mathcal{L}_{\text{total}}$. The relative improvement of t$\mathcal{L}_{\text{total}}$ is about 4.45% in the case of 'original ELBO'.

## 5.3 EVALUATION: MOLECULAR GENERATION PHASE

The generated molecules from the generative model can be evaluated according to three criteria. The first one is a validity. This means that the generated molecule has a sound structure. This can be

Table 3: Generative efficiency at each property (ZINC310K)

|  | molwt | LogP | QED | average |
|---|---|---|---|---|
| **ssVAE** (Kang & Cho, 2018) | 0.760 | 0.0.866 | 0.901 | 0.842 |
| vanilla **VAE** | 0.800 | 0.868 | 0.880 | 0.849 |
| **LOVAE** | **0.897** | **0.924** | **0.937** | **0.920** |

Table 4: Probability that the property value of the generated molecule falls within 10% and 5% of the error range of the condition value (ZINC310K)

|  | **vanilla VAE** | | | | **LOVAE** | | | |
|---|---|---|---|---|---|---|---|---|
|  | molwt | LogP | QED | average | molwt | LogP | QED | average |
| within 10% | 0.791 | 0.577 | 0.616 | 0.662 | 0.893 | 0.703 | 0.630 | **0.742** |
| within 5% | 0.789 | 0.377 | 0.413 | 0.526 | 0.879 | 0.450 | 0.414 | **0.581** |

determined using the RDKit package (Landrum). The second one is novelty. The purpose of the inverse molecular design is to find new molecules that have not discovered yet. A molecule is said to be novel if it is not in the train set. The third one is uniqueness. If the latent space is very narrow and latent variable $z$ is repeatedly sampled in a similar space, there is much room for generating the same SMILES. That is, the more uniqueness, the better the generative model. The generated molecules are more meaningful if they satisfy all three criteria, validity, novelty, and uniqueness. In this paper, the ratio that satisfies all three criteria is defined as a generative efficiency. For example, if a generative model attempts to generate 1,000 molecules and has 600 molecules that satisfy validity, novelty, and uniqueness at once, the generative efficiency is 0.6.

For evaluation, three values were determined for each property as a condition for the generative model, and ZINC310K was used. It was determined to be close to the median, lower 10%, and upper 10% value in our train set. In case of molwt, 360.0, 260.0, and 460.0 were used as the condition. For LogP and QED, {3.0, 1.5, 4.5} and {0.7, 0.5, 0.9} were chosen, respectively. The molecular generation was attempted 3,500 times in each condition value, and 31,500 molecules were generated in a total of 9 conditions. For the analysis, 'original ELBO' models were used. The results of the generative efficiency are shown in Table 3. It can be seen that LOVAE has good generative efficiency and uniform performance in all properties.

## 5.4 EVALUATION: PROPERTY SATISFACTION

In addition to the generative efficiency, it is possible to use property satisfaction as a measure of performance for cVAE. It is how many molecules with properties being close to the target condition can be generated. For this, evaluation of property satisfaction was conducted based on two criteria. The first is the percentage that the property value of the generated molecule falls within 10% of the error range of the condition value. For example, if the target value of molwt is 360.0, it measures the percentage of the generated molecules whose property value lies between 324.0 and 396.0. The results of the experiment are shown in Table 4. In all three properties, LOVAE showed higher property satisfaction. At 10% and 5% error property satisfaction, LOVAE showed a relatively 12.1% and 10.5% improvement, respectively.

For the comparison with previous works using ZINC250K, we referred to You et al. (2018b). In that paper, property targeting task was performed, and specific ranges of molwt and LogP were considered. In our case, if the target range of molWt is from 150 to 200, we conditioned LOVAE as 175. The target ranges are four like Table 5.

Except for the target range -2.5 $\leqq$ LogP $\leqq$ -2.0, LOVAE showed the best performance. Since percentages of the training data in the range of -2.5 $\leqq$ LogP $\leqq$ -2.0 and 5 $\leqq$ LogP $\leqq$ 5.5 are 0.28% and 1.30%, the first target range can be a bit more difficult. In this respect, the result of LOVAE at the target range -2.5 $\leqq$ LogP $\leqq$ -2.0 seems to make sense.

Table 5: Probability that the property value of the generated molecule falls within target range (ZINC250K)

| target range | LogP | | molwt | |
|---|---|---|---|---|
| | $-2.5 \sim -2.0$ | $5.0 \sim 5.5$ | $150 \sim 200$ | $500 \sim 550$ |
| **JT-VAE** (Jin et al., 2018) | 0.113 | 0.076 | 0.007 | 0.160 |
| **ORGAN** (Guimaraes et al., 2017) | 0.000 | 0.002 | 0.151 | 0.001 |
| **GCPN** (You et al., 2018a) | **0.855** | 0.547 | 0.761 | 0.741 |
| **LOVAE** | 0.316 | **0.606** | **0.992** | **0.976** |

Table 6: Properties of the top three optimized molcules trained on ZINC250K

| | penalized LogP | | | QED | | |
|---|---|---|---|---|---|---|
| | Top 1 | Top 2 | Top 3 | Top 1 | Top 2 | Top 3 |
| **GrammarVAE** (Kusner et al., 2017) | 2.94 | 2.89 | 2.80 | - | - | - |
| **RevalancingVAE** (Yan et al., 2019) | 5.32 | 5.28 | 5.23 | - | - | - |
| **GCPN** (You et al., 2018a) | 7.98 | 7.85 | 7.80 | 0.948 | 0.947 | 0.946 |
| **JT-VAE** (Jin et al., 2018) | 5.30 | 4.93 | 4.49 | 0.925 | 0.911 | 0.910 |
| **AllSMILES** (Alperstein et al., 2019) | 16.42 | 16.32 | 16.21 | 0.948 | 0.948 | 0.948 |
| **molecularRNN** (Popova et al., 2019) | 10.34 | 10.19 | 10.14 | 0.948 | 0.948 | 0.947 |
| **graphAF** (Shi et al., 2020) | 12.23 | 11.29 | 11.5 | 0.948 | 0.948 | 0.948 |
| **LOVAE** | **20.59** | **18.26** | **16.39** | 0.948 | 0.948 | 0.948 |
| **LOVAE** with plogP | 15.47 | 15.46 | 15.45 | - | - | - |

## 5.5 EVALUATION: PROPERTY MAXIMIZATION (PENALIZED LOGP AND QED)

In the many previous papers, a property maximization task was performed and it was evaluated on penalized LogP (pLogP) and QED (Kusner et al., 2017). QED is a property with a boundary range with [0, 1], but the range of penalized logP is $(-\infty, \infty)$. pLogP is a LogP penalized by the synthetic accessibility score (SA) and the number of large rings (cycle), pLogP = logP - SA - cycle. It can be thought as a extrapolation task because generative models have to create a new molecule that is not in the range of the property values of the training DB (The maximum pLogP in training DB = 5.072). In order to find new molecules which show the highest property, some of previous approaches utilized a reward function or a property regressors with sparse Gaussian process (Kusner et al., 2017; Shi et al., 2020; Gómez-Bombarelli et al., 2018). In our approach, LOVAE, we just conditioned by a high value such as 30.0 and 0.98 for LogP and QED, respectively.

Table 6 shows the property maximization results. LOVAE generated the new molecules shown the highest penalized LogP and QED property. It is noteworthy that LOVAE showed good performance only with LogP condition without a separate part like a reward or property regressor. That is, it was confirmed that LOVAE, which is a conditional VAE type, works properly even in the extrapolation task. Top 3 molecules of each property are represented Fig. 4. In addition, LOVAE with pLogP as condition was trained and verified. The performance was worse than LOVAE with LogP, but since the number of large rings can also be given as condition, the trend of the generated molecules was different (5).

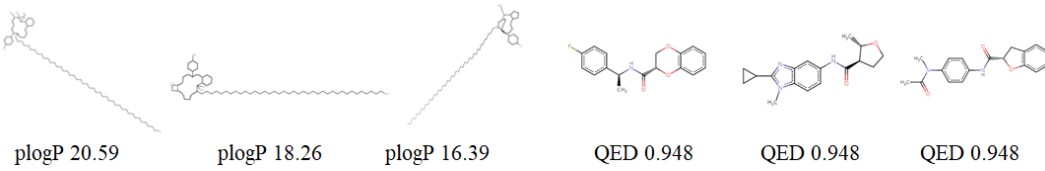

| plogP 20.59 | plogP 18.26 | plogP 16.39 | QED 0.948 | QED 0.948 | QED 0.948 |

Figure 4: Samples of generated molecules of LOVAE

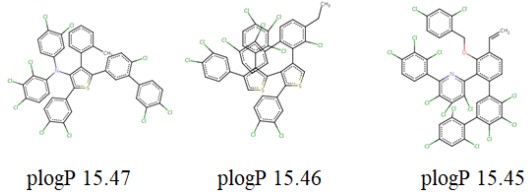

plogP 15.47        plogP 15.46        plogP 15.45

Figure 5: Samples of generated molecules of LOVAE with condition of plogP

## 6    CONCLUSION

In this paper, LOVAE applying latent optimization technique to VAE algorithm is proposed. By training the encoder first, the latent variable $\mathbf{z}$ has an updated distribution in the direction of reducing the train loss. Additionally, $\mathbf{z}$ is updated before training the decoder in the direction of reducing the reconstruction loss. The training of decoder becomes more efficient by utilizing the optimized $\mathbf{z}$. This was applied and verified in the inverse molecular design task in ZINC dataset, and confirmed that it showed a better appearance in the train loss, ELBO, and molecular generative performance than those of vanilla VAE.

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
