# OpenReview forum: "LATENT OPTIMIZATION VARIATIONAL AUTOENCODER FOR CONDITIONAL MOLECULAR GENERATION"
_ICLR.cc/2021/Conference — Reject_

### Official Review · AnonReviewer4 · 2020-10-28

**Rating:** 4
**Confidence:** 4

**Review:**


The paper proposes a latent optimization VAE (LOVAE) to insert an optimization step of the latent variable and alternately update the encoder and decoder of conditional VAE. The additional latent optimization step optimizes phi with respect to the reconstruction loss. It makes sense the reconstruction loss is low in table 1.


1: Although the adaptation of latent optimization to cVAE is interesting and can improve the performance, the idea or the contribution is not significant enough, especially considering many existing VAE variants (including those already discussed in the paper).

2. One of the key results -- to generate molecules with similar property as the conditional property. Table 3 only compares with vanilla VAE, which doesn’t take condition property as input. The baseline is not persuasive. It will be more convincing if the authors can compare with other VAE types (i.e. include, but not limited to cVAE) and other non-VAE but reverse molecule design methods (i.e. GAN type, or not generative models).  Additionally, it is less convincing that the authors compare with different sets of existing methods on different tasks, for example Table 3, Table 4, Table 5. What are the results look like if say we add Table 4's other methods as a comparison for  table 3?  for example. JT-VAE (Jin et al., 2018)  , ORGAN (Guimaraes et al., 2017)  ,GCPN (You et al., 2018a) . The paper may need some more coherent experiments to be more persuasive.



3. It will be more clear if the authors can mark the best values in tables in bold.

---

> ### Author Response · Authors · 2020-11-13
> **Response part 1**
>
> Comment 1: The idea or the contribution is not significant enough, especially considering many existing VAE variants (including those already discussed in the paper).
>
> Response 1:
> First of all, thank you for pointing out the good points. That’s all I agree with. Our paper began to develop VAE training in terms of a latent variable optimization. The problem we originally wanted to consider in this paper is that vanilla VAE training is not in an optimized state. A decoder is trained depending on an encoder output. However, in the vanilla VAE, the latent value (z), which is the result of encoder before the update, is used for the decoder. From the perspective of the decoder, when the same input x is used, it may be more effective to calculate the loss by using the latent variable (z’) passing through the updated encoder. Our proposed method, LOVAE, tried to solve this problem in terms of the latent variable optimization. LOVAE uses the same input x for learning the decoder and encoder, and z’ can be used for updating the decoder because the encoder is updated first, so the loss becomes smaller than the vanilla VAE. In addition, it helped the decoder training by optimizing z’ in the direction of reducing the loss(z’) one more in a way that does not spoil the training of encoder and decoder. This can be seen in Table 1.
>
> There are cases where encoder is considered first like SA-VAE [1], but this does not update encoder first. After simply updating z in the direction of reducing the decoder loss dozens of times, encoder and decoder are separately trained by this final z. In the case of [2], encoder is updated several times until certain conditions are satisfied, and different inputs x are used when updating encoder and decoder. LOVAE is similar to the previous two methods, but differs in some ways. It updates the encoder first, but updates it only once, and uses the same input x for the encoder and decoder, and performs additive latent optimization to help overall VAE learning. We think z’ obtained by updating the encoder at the same input x is more natural and suitable for learning the decoder than SA-VAE’s final latent variable. In addition, we think that using the same input x for the training encoder and decoder is more effective in terms of latent optimization than using different input x to the encoder and decoder.
>
> Although LOVAE was applied to different DB and task from the above papers, we compared some part to [1] and [2]. Compared to vanilla VAE, we checked how much the loss was relatively reduced. (We used the Yahoo dataset results of [1] and [2])
>
> LOVAE: 4.45 % / Lagging [1]: 0.24 % / SA-VAE [2]: 0.82 %
>
> As shown in the table above, LOVAE showed the largest loss reduction (or ELBO increase). Also, the training time was compared with reference to [1]. (We used the Yahoo dataset results of [1] and [2])
>
> vanilla VAE: 1.00 / LOVAE: 1.75 / Lagging [1]: 2.2 / SA-VAE [2]: 9.91
>
> Considering the time of vanilla VAE as 1.00, LOVAE takes 1.75 times as much time. LOVAE is expected to show faster learning speed than [1]. Although the DB is different, it seems that the comparison is possible because it was compared to the same vanilla VAE method.
>
> We will clarify the connection between the proposed technology and the problem we are trying to solve. In addition, as above, the proposed method will be more clarified and a description of the existing method will be added.
>
> [1] Junxian He et al. Lagging Inference networks and posterior collapse in Variational Autoencoders (ICLR 2019)
>
> [2] Yoon Kim et al. Semi-amortized variational autoencoders. International Conference on Machine Learning (ICML), 2018b.
>
>
> Comment 2: Table 3 only compares with vanilla VAE, which doesn’t take condition property as input.
>
> Response 2:
> Sorry for the confusion. All vanilla VAE, including Table 3, are conditional VAE. In other words, the property value was used as condition in the vanilla VAE as well.
>
>
> Comment 3: It will be more clear if the authors can mark the best values in tables in bold.
>
> Response 3:
> Thanks for the point. I'll fix it right away.

---

> > ### Comment · AnonReviewer4 · 2020-11-25
> > **thank you for the reply**
> >
> > Thank you for the reply and new experiments.  Can you please explain more my second point, especially "comparing with different sets of existing methods on different tasks"?

---

> > > ### Author Response · Authors · 2020-11-25
> > > **Reasons for using different sets**
> > >
> > > First of all, sorry for not giving a clear answer to your previous question. There are several verification databases in the molecular field, such as ZINC DB and QM9. Among them, ZINC DB was chosen because of its larger molecular size than QM9. Also, ZINC DB is made up of hundreds of millions of molecules. So, in existing papers, tens of thousands of samples are sampled and verified. Among them, existing papers use the 250k version [1] or the 310k version [2]. There may be a problem with reliability when I implement and verify all existing papers. So, the proposed method LOVAE was verified with the environment mentioned in the same existing paper in both sets.
> > >
> > >
> > > [1] Matt J Kusner, Brooks Paige, and José Miguel Hernández-Lobato. Grammar variational autoencoder. In Proceedings of the 34th International Conference on Machine Learning-Volume 70, pp. 1945– 1954. JMLR. org, 2017.
> > >
> > > [2] Chaochao Yan, ShengWang, Jinyu Yang, Tingyang Xu, and Junzhou Huang. Re-balancing variational autoencoder loss for molecule sequence generation. arXiv preprint arXiv:1910.00698, 2019.

---

### Official Review · AnonReviewer1 · 2020-10-28
**Application of existing work**

**Rating:** 5
**Confidence:** 4

**Review:**

The authors propose extensions to VAE training to stabilize the learning process. First, the encoder is trained while the decoder is fixed. This is followed by an additive optimization of the latent variable from the encoder. Finally, the decoder is updated based on the updated latent variable from the encoder. They conduct experiments on molecule generation tasks to show the efficacy of their method. The paper is well written and easy to follow.

My main concern with this paper is novelty. Their method of pretraining the encoder is not particularly novel and has been applied in Natural Language Processing. For example, in He et al (ICLR 2019) [2], the encoder and decoder steps of the VAE are separated and more steps of the encoder are performed in each iteration.
Also in  another related work, Li et al (EMNLP 2019), they combine this method with another trick (KL-thresholding). The difference between the proposed method and these existing methods are not clear (and I believe are minor). In fact the method of He at al, can be considered as a generalization of the proposed method, with minor differences.

If most of the techniques proposed in the paper are well known in the NLP community, then the only novelty is the application of this technique to SMILES strings. It is not surprising at all that it is better than vanilla VAE training. Even in this case, I think the comparisons with existing work is not thorough.

Is it possible to compare the generated molecules on a broader set of criteria, for example using the MOSES evaluation framework? [1]. This allows one to evaluate the efficacy of the method along different criteria like validity, novelty, uniqueness, FCD, etc and compare it with a broader set of methods. The code is available at https://github.com/molecularsets/moses

I would request the authors to carefully position their work with related work mentioned above.

[1] Polykovskiy et al. Molecular Sets (MOSES): A Benchmarking Platform for Molecular Generation Models

[2] He et al. Lagging Inference networks and posterior collapse in Variational Autoencoders (ICLR 2019)

[3] Li et al. A Surprisingly Effective Fix for Deep Latent Variable Modeling of Text (EMNLP 2019)

---

> ### Author Response · Authors · 2020-11-13
> **Response part 1**
>
> Comment 1:  Their method of pretraining the encoder is not particularly novel and has been applied in Natural Language Processing. For example, in He et al (ICLR 2019) [2], the encoder and decoder steps of the VAE are separated and more steps of the encoder are performed in each iteration.
>
> Response 1:
> Thank you for sharing this paper. The analysis on VAE seems to be very good. It is true that our proposed method (LOVAE) was used for the reasons pointed out in [1]. But, LOVAE focused on a latent variable optimization, and it differs in three ways. First, LOVAE performs the encoder update only once without a stop condition. It is expected to be faster training. Second, LOVAE uses the same input x to learn the encoder and decoder. In [1], the encoder was updated several times with sampled input x, and then the decoder was trained with newly sampled input x’. In the case of LOVAE, after updating the encoder, we update the decoder using z' from the same x. The z’ from the updated encoder has a lower decoder loss than the z from the non-updated encoder. (loss(z) > loss(z’)) If it is a new input x’ like [1], it is information that was not reflected when the encoder was updated, and z obtained in this way may have little latent optimization effect. Third, the additive optimization in LOVAE helped the decoder training by optimizing z’ in the direction of reducing the loss(z’) one more in a way that does not spoil the decoder training. This can be seen in Table 1.
>
> Although LOVAE was applied to different DB and task from the above papers, we compared some part to [1] and [2]. Compared to vanilla VAE, we checked how much the loss was relatively reduced. (We used the Yahoo dataset results of [1] and [2])
>
> LOVAE: 4.45 % / Lagging [1]: 0.24 % / SA-VAE [2]: 0.82 %
>
> As shown in the table above, LOVAE showed the largest loss reduction (or ELBO increase). Also, the training time was compared with reference to [1]. (We used the Yahoo dataset results of [1] and [2])
>
> vanilla VAE: 1.00 / LOVAE: 1.75 / Lagging [1]: 2.2 / SA-VAE [2]: 9.91
>
> Considering the time of vanilla VAE as 1.00, LOVAE takes 1.75 times as much time. LOVAE is expected to show faster learning speed than [1]. Although the DB is different, it seems that the comparison is possible because it was compared to the same vanilla VAE method.
>
> When revising this paper, I will refer to [1] paper and mention the differences.
>
> [1] Junxian He et al. Lagging Inference networks and posterior collapse in Variational Autoencoders (ICLR 2019)
>
> [2] Yoon Kim et al. Semi-amortized variational autoencoders. International Conference on Machine Learning (ICML), 2018b.
>
>
> Comment 2: for example using the MOSES evaluation framework?
>
> Response 2:
> I agree with the comment you gave. While doing this task, I was contemplating which measurement was appropriate. Like the comments you gave, uniqueness and novelty may be lacking when analyzing performance. I will check the MOSES you informed.

---

> > ### Comment · AnonReviewer1 · 2020-11-25
> > **Thank you for the new experiments**
> >
> > Thank you for the detailed response and new experiments. Please update the MOSES evaluation if you are able to.

---

### Official Review · AnonReviewer2 · 2020-10-29
**Motivation and technical content unclear - reject**

**Rating:** 3
**Confidence:** 4

**Review:**

The paper proposes a way to use conditional variational autoencoder setups to design materials conditioned on properties (labels). The problem is of some significance in the materials design space using generative models, using machinery from neural network modeling - RNNs, VAEs, GANs and optimization.

The writing is clear, and material is readable. However, I note the following issues with the content:

1. Problem motivation not very clear
I felt that the paper does not provide good enough context into the problem and had to piece it together from related work cited (e.g. [1], [2]). In particular, the work seems to be quite related to the work [1] by Kang and Cho, where they use a graphical model fashioned as a VAE, with the dependencies being the label or property y, the latent variable z and the molecule x - i.e. ELBO(log p(x, y)) in equation (2) of the paper.

The paper's claim is that they address problems with the vanilla conditional VAE setup. However, it seemed difficult to figure out what exactly the paper was trying to fix in the original VAE setup. They mention "posterior collapse", although there does not seem to be much evidence given (please clarify) to show these deficiencies in the setup - I assume that we should look at section 5.2, Table 1, but no mention of posterior collapse is made in these likelihood numbers.

I would also like to quote the following lines from the beginning of section 4. It is unclear again what is meant by saying that the decoder is sensitive to the encoder results and the accuracy might vary ...

"Since the latent variable z generated from the encoder is inputted to the decoder to calculate Ltotal,
the encoder and decoder are closely related. If the encoder generates non-informative or highly
volatile z, then the decoder is difficult to reconstruct x from z. In this case, it may not be optimal to
update the encoder and decoder simultaneously. The decoder is very sensitive to the encoder results,
z, and the accuracy of the decoder training may vary according to the encoder results. In other words,
the result of the encoder can be optimized in advance in direction in which the decoder trains well.
That is, better encoder and better latent variable can make the decoder even better."

2. The optimization process: Aside from the issues regarding the paper's motivation, the optimization procedure adopted does not seem to be very clear and could perhaps need clarification. The paper mentions a 'two step' optimization process, of first updating the encoder (keeping decoder fixed) and then updating the decoder. This is strongly suggestive of the wake-sleep algorithm by Hinton et al [2], but if that is so, the exposition would need considerable elaboration with equations, optimization objectives, and clarification on how the optimization machinery used solves the actual problems at hand (which I also had some difficulty following, as explained in the previous point). In other words, the connection between the technique adopted and the problem solved is not clear.

I think the paper needs additional work in explaining the motivation of the problem with examples on why the original VAE setup is deficient and what the proposed fixes do to solve the problem at hand.

[1] Kang and Cho: https://arxiv.org/pdf/1805.00108.pdf

[2] Wake-sleep algorithm https://www.cs.toronto.edu/~hinton/csc2535/readings/ws.pdf

---

> ### Author Response · Authors · 2020-11-13
> **Response part 1**
>
> First of all, thank you for pointing out the good points. That’s all I agree with. I will answer all of your comments.
>
> What you pointed out is correct. Pointing out the main problem of our paper as “post-collapse” seems to have been misleading. One of the main problems with VAE is post-collapse, which was fortunately not seen in the vanilla VAE experiment in our paper (sequential representation of molecules). The problem we originally wanted to consider in this paper is that vanilla VAE training is not in an optimized state. A decoder is trained depending on an encoder output. However, in the vanilla VAE, the latent value (z), which is the result of the encoder before the update, is used for the decoder. From the perspective of the decoder, when the same input x is used, it may be more effective to calculate the loss by using a latent variable (z’) passing through the updated encoder.
>
> Our proposed method, LOVAE, tried to solve this problem in terms of latent variable optimization. LOVAE uses the same input x for learning decoder and encoder, and z’ is used for learning the decoder because the encoder is updated first, so the loss becomes smaller than the vanilla VAE. In addition, it helped the decoder training by optimizing z’ in the direction of reducing the loss(z’) one more in a way that does not spoil the decoder training. This can be seen in Table 1.
>
> There are cases where encoder is considered first like SA-VAE [1], but this does not update encoder first. After simply updating z in the direction of reducing the decoder loss dozens of times, the encoder and decoder are separately trained by this final z. In the case of [2], encoder is updated several times until certain conditions are satisfied, and different inputs x are used when updating encoder and decoder. LOVAE is similar to the previous two methods, but differs in some ways. It updates the encoder first, but updates it only once, and uses the same input x for the encoder and decoder, and performs additive latent optimization to help overall VAE learning. We think z’ obtained by updating the encoder at the same input x is more natural and suitable for learning the decoder than SA-VAE’s final latent variable. In addition, we think that using the same input x for the training encoder and decoder is more effective in terms of latent optimization than using different input x to the encoder and decoder.
>
> Although LOVAE was applied to different DB and task from the above papers, we compared some part to [1] and [2]. Compared to vanilla VAE, we checked how much the loss was relatively reduced. (We used the Yahoo dataset results of [1] and [2])
>
> LOVAE: 4.45 % / Lagging [1]: 0.24 % / SA-VAE [2]: 0.82 %
>
> As shown in the table above, LOVAE showed the largest loss reduction (or ELBO increase). Also, the training time was compared with reference to [1]. (We used the Yahoo dataset results of [1] and [2])
>
> vanilla VAE: 1.00 / LOVAE: 1.75 / Lagging [1]: 2.2 / SA-VAE [2]: 9.91
>
> Considering the time of vanilla VAE as 1.00, LOVAE takes 1.75 times as much time. LOVAE is expected to show faster learning speed than [1]. Although the DB is different, it seems that the comparison is possible because it was compared to the same vanilla VAE method.
>
> We will clarify the connection between the proposed technology and the problem we are trying to solve. In addition, as above, the proposed method will be more clarified and a description of the existing method will be added.
>
> [1] Junxian He et al. Lagging Inference networks and posterior collapse in Variational Autoencoders (ICLR 2019)
>
> [2] Yoon Kim et al. Semi-amortized variational autoencoders. International Conference on Machine Learning (ICML), 2018b.

---

> > ### Comment · AnonReviewer2 · 2020-11-25
> > **Thanks for the clarifications**
> >
> > I have read the author responses and reviewer comments. The references clarify my concerns about the optimization process somewhat. I will still to my score though as it seems to me that we ought to explain this in more detail in the paper.

---

### Official Review · AnonReviewer3 · 2020-11-03
**LATENT OPTIMIZATION VARIATIONAL AUTOENCODER FOR CONDITIONAL MOLECULAR GENERATION**

**Rating:** 4
**Confidence:** 2

**Review:**

The authors propose a novel way to train variational auto-encoders and apply it to generate small molecules. The paper is well presented and the method easy to follow. Overall I think the goals of the article are somewhat disjoint, and it feels like the proposed method (LOVAE) should be applicable to standard datasets as well (e.g. CIFAR10), which would have helped the “story” of the paper. This is especially important since I find the motivation for the LOVAE reasonably weak, and there’s not much in terms of theoretical backing.

Some further comments are as follows:
* Glow isn’t really a good “general reference” on reversible generative models, many successful versions exist prior to it.
* The QED metric seems to top out at 0.948, why does this happen? It doesn’t look very informative as is.
* All images, notably Figure 5, should be in vector format. It’s currently hard to see what’s in there and it does not scale.
* There are some language errors, e.g. “the training of decoder”. It might be worth having another look at language throughout.

---

> ### Author Response · Authors · 2020-11-13
> **Response part 1**
>
> Comment 1: the proposed method (LOVAE) should be applicable to standard datasets as well (e.g. CIFAR10), which would have helped the “story” of the paper.
>
> Response 1: Thanks for the good point. As you said, the proposed method can be used for various domain data. VAE was a bit more difficult to apply in text with sequential characteristics. In addition, there have been many studies on the molecular generation based on VAE in recent years, and since SMILES can also be viewed as a kind of text, I thought it would be appropriate for verification. In addition, the task of creating a new molecule that does not exist was more attractive than the task of simply creating a sentence. However, if we apply our proposed method to the existing task (ex. image, text) and show the result, it will be a better study as you said.
>
>
> Comment 2:  This is especially important since I find the motivation for the LOVAE reasonably weak, and there’s not much in terms of theoretical backing.
>
> Response 2:
> First of all, thank you for pointing out the good points. That’s all I agree with. Our paper began to develop VAE training in terms of a latent variable optimization. The problem we originally wanted to consider in this paper is that vanilla VAE training is not in an optimized state. A decoder is trained depending on an encoder output. However, in the vanilla VAE, the latent value (z), which is the result of encoder before the update, is used for the decoder. From the perspective of the decoder, when the same input x is used, it may be more effective to calculate the loss by using the latent variable (z’) passing through the updated encoder. Our proposed method, LOVAE, tried to solve this problem in terms of the latent variable optimization. LOVAE uses the same input x for learning the decoder and encoder, and z’ can be used for updating the decoder because the encoder is updated first, so the loss becomes smaller than the vanilla VAE. In addition, it helped the decoder training by optimizing z’ in the direction of reducing the loss(z’) one more in a way that does not spoil the training of encoder and decoder. This can be seen in Table 1.
>
> There are cases where encoder is considered first like SA-VAE [1], but this does not update encoder first. After simply updating z in the direction of reducing the decoder loss dozens of times, encoder and decoder are separately trained by this final z. In the case of [2], encoder is updated several times until certain conditions are satisfied, and different inputs x are used when updating encoder and decoder. LOVAE is similar to the previous two methods, but differs in some ways. It updates the encoder first, but updates it only once, and uses the same input x for the encoder and decoder, and performs additive latent optimization to help overall VAE learning. We think z’ obtained by updating the encoder at the same input x is more natural and suitable for learning the decoder than SA-VAE’s final latent variable. In addition, we think that using the same input x for the training encoder and decoder is more effective in terms of latent optimization than using different input x to the encoder and decoder.
>
> Although LOVAE was applied to different DB and task from the above papers, we compared some part to [1] and [2]. Compared to vanilla VAE, we checked how much the loss was relatively reduced. (We used the Yahoo dataset results of [1] and [2])
>
>    LOVAE: 4.45 %  / Lagging [1]: 0.24 %  / SA-VAE [2]: 0.82 %
>
> As shown in the table above, LOVAE showed the largest loss reduction (or ELBO increase). Also, the training time was compared with reference to [1]. (We used the Yahoo dataset results of [1] and [2])
>
>    vanilla VAE: 1.00  / LOVAE: 1.75  / Lagging [1]: 2.2  / SA-VAE [2]: 9.91
>
> Considering the time of vanilla VAE as 1.00, LOVAE takes 1.75 times as much time. LOVAE is expected to show faster learning speed than [1]. Although the DB is different, it seems that the comparison is possible because it was compared to the same vanilla VAE method.
> We will clarify the connection between the proposed technology and the problem we are trying to solve. In addition, as above, the proposed method will be more clarified and a description of the existing method will be added.
>
> [1] Junxian He et al. Lagging Inference networks and posterior collapse in Variational Autoencoders (ICLR 2019)
> [2] Yoon Kim et al. Semi-amortized variational autoencoders. International Conference on Machine Learning (ICML), 2018b.

---

> ### Author Response · Authors · 2020-11-13
> **Response part 2**
>
> Comment 3: Glow isn’t really a good “general reference” on reversible generative models, many successful versions exist prior to it.
>
> Response 3 :
> Thank you for pointing out that. I agree with your comment. We will add an appropriate existing study as below as a reference.
>
>
> Comment 4: The QED metric seems to top out at 0.948, why does this happen? It doesn’t look very informative as is.
>
> Response 4:
> Sorry for not putting in a detailed description. QED (Quantitative Estimate of Druglikeness) stands for quantitative estimation of drug-likeness and the concept has been introduced by Richard Bickerton and coworkers [1]. QED is a property with a boundary range. The range of penalized logP is (−∞,∞), while the range of QED is [0, 1]. The highest QED from RDkit tool is 0.948. I will add this to the paper.
>
> [1] Bickerton, G.R.; Paolini, G.V.; Besnard, J. Muresan, S. Hopkins, A.L. ‘Quantifying the chemical beauty of drugs’, Nature Chemistry, 4, 90-98, 2012
>
>
> Comment 5: All images, notably Figure 5, should be in vector format. It’s currently hard to see what’s in there and it does not scale.
>
> Response 5:
> Yes, I will modify it to vector format as you said. Thank you for pointing out.
>
>
> Comment 6: here are some language errors, e.g. “the training of decoder”. It might be worth having another look at language throughout.
>
> Response 6:
> The language expression will be revised as a whole. Thank you.

---

### Decision · Program_Chairs · 2021-01-07
**Final Decision**

**Decision:**

Reject

**Comment:**

The reviewers agreed that the paper can be improved in several aspects such a motivation, novelty and framing of the contribution. The reviewers also liked the clarity of the presentation and some of the ideas. The reviewers constructive input may be used to improve this work.